# No Time to Die—How Islets Meet Their Demise in Transplantation

**DOI:** 10.3390/cells12050796

**Published:** 2023-03-03

**Authors:** Atharva Kale, Natasha M. Rogers

**Affiliations:** 1Centre for Transplant and Renal Research, Westmead Institute for Medical Research, Westmead, NSW 2145, Australia; 2Faculty of Medicine and Health, University of Sydney, Sydney, NSW 2006, Australia; 3Renal and Transplant Unit, Westmead Hospital, Westmead, NSW 2145, Australia

**Keywords:** islet transplantation, beta cells, rejection, IBMIR, ER stress

## Abstract

Islet transplantation represents an effective treatment for patients with type 1 diabetes mellitus (T1DM) and severe hypoglycaemia unawareness, capable of circumventing impaired counterregulatory pathways that no longer provide protection against low blood glucose levels. The additional beneficial effect of normalizing metabolic glycaemic control is the minimisation of further complications related to T1DM and insulin administration. However, patients require allogeneic islets from up to three donors, and the long-term insulin independence is inferior to that achieved with solid organ (whole pancreas) transplantation. This is likely due to the fragility of islets caused by the isolation process, innate immune responses following portal infusion, auto- and allo-immune-mediated destruction and β-cell exhaustion following transplantation. This review covers the specific challenges related to islet vulnerability and dysfunction that affect long-term cell survival following transplantation.

## 1. Introduction

Diabetes mellitus (DM) is a developing global health emergency. Current estimates suggest that >500 million people worldwide are affected, with an increasing prevalence in low- and middle-income countries. DM, regardless of aetiology, has a unique capacity to affect multiple organ systems that predispose to a substantially increased risk of cardiovascular disease, chronic kidney and liver diseases, malignancy and neurological impairment. The long-term economic burden that accompanies a diagnosis of DM and the development of complications create concerns about the cost and utilisation of healthcare resources over the duration of the disease. Type 1 DM (T1DM) is characterised by the autoimmune destruction of β-cells, although the notion of β-cell complicity in their own destruction [1] due to limited responses to survive an inflammatory insult is touted as a competing hypothesis for disease development. Type 2 DM (T2DM) is caused by a combination of β-cell dysfunction and insulin resistance [2]. The failure of insulin-sensitive tissues to respond appropriately to insulin leads to compensatory hyperinsulinemia, which facilitates β-cell dysfunction and death through exhaustion. Islet transplantation is an accepted treatment option for patients with T1DM [3]—and not yet for patients with T2DM primarily due to a lack of sufficient donors—but despite the initial success in reducing hypoglycaemia unawareness, long-term allograft survival and insulin independence are limited [4] due to an inexorable decline in β-cell number and function. The loss of critical islet mass is common in both type 1 and type 2 DM, as well as in islet transplantation. Our understanding of islet biology, particularly how these cells die in response to exogenous stressors, remains key to developing novel treatments that protect the endocrine pancreas and facilitate the survival of transplanted β-cells.

## 2. Mechanisms of Islet Cell Death in Islet Transplantation

Insulin replacement has remained the standard management for patients with T1DM and for patients with late-stage T2DM. However, it is not able to provide complete physiological metabolic control. Allogeneic islet cell transplantation remains an effective treatment for patients with T1DM who have concurrent hypoglycaemia unawareness and metabolic instability [3]. Hypoglycaemia can pass as unrecognised in a subset of patients with diabetes—this condition is known as hypoglycaemic unawareness—where the development of neuroglycopenia is not preceded by autonomic warning symptoms (e.g., tremors and sweating) due to a loss of sympathetic and adreno-medullary counterregulatory mechanisms, as well as a lack of α-cell responsiveness [5,6]. The development of hypoglycaemia unawareness is associated with the duration of diabetes in the context of tight metabolic control. It is most commonly observed in patients with T1DM, affecting 30–40% of patients, compared to patients with insulin-dependent T2DM [7]. Hypoglycaemic unawareness is associated with the development of severe hypoglycaemia leading to increased morbidity and mortality [8,9].

Islet transplantation provides the replacement of β-cell (and potentially α-cell) function, and it has been shown to effectively reduce hypoglycaemia unawareness, improve quality of life and provide durable insulin independence [3,10]. Islet transplantation also slows the micro- and macro-vascular complications that impact morbidity and mortality [11], as well as healthcare resource use, although the procedure is not necessarily cost-effective [12]. The ability to apply this therapy to a broader range of patients with diabetes is also limited by donor pancreas availability. Patients typically require three separate transplantation procedures to maximise the efficacy and stability of islet cell mass. The majority of islets are thought to be lost in the immediate peri-transplant period following infusion and engraftment. This is typically followed by a second phase of cellular loss, where an inexorable decline in β-cell function can be mediated by allo- and auto-immune destruction, as well as by non-immunological functional impairment and destruction that resembles the failure of islets in T2DM (inflammatory, ER amyloid and metabolic stressors, Figure 1) [13]. This results in long-term insulin independence rates of <30% [14].

## 3. Instant Blood Inflammatory Reaction (IBMIR) and Early Islet Demise

Inflammation during the early stages of islet transplantation has been identified as one of the major reasons for poor long-term graft survival. Approximately 25% of transplanted islets are lost immediately [15] as they come into contact with intraportal ABO-compatible whole blood, triggered by exposed tissue factor on the surface of the islets [16]. Tissue factor interacts with factor VIIa to activate the extrinsic coagulation pathway. IBMIR is therefore characterised by coagulation, complement activation, the recruitment and infiltration of leukocytes and the production of proinflammatory cytokines/chemokines, all of which lead to the regulated necrosis [17] of β-cells. Concurrent increases in thrombin–antithrombin levels and c-peptide are indicative of islet damage and cell lysis, although this can be mitigated by the use of heparin [18]. Extra-hepatic transplantation sites have been considered alternative locations for islet deposition, as they eliminate the potential for IBMIR, and while they have been effective in small animal models, these have typically produced poorer clinical outcomes [19]. The use of encapsulation technology as a cytoprotective mechanism has been extensively explored (well-reviewed in [20]) but has not been robustly translated to clinical practice.

The study of IBMIR in vivo is difficult to recapitulate due to multiple interacting components; however, whole blood models of human allo-islet transplantation have been developed, and the early (<6 h) innate immune response has been well characterised. A recent study investigated IBMIR for up to 48 h [21], demonstrating ongoing thrombin–antithrombin complexes and platelet activation at 12 h post-transplantation. This was accompanied by increased expressions of chemokines, including interferon-inducible T-cell chemoattractant (the CXCR3 ligand), soluble CD40 ligand and monocyte chemoattractant protein-1 (the CCR2 ligand), and a massive infiltration of neutrophils and monocytes.

Natural killer (NK) cells and macrophages also contribute to IBMIR. The liver—a typical site of β-cell deposition following intraportal infusion—and its substantial mononuclear cell population contain a large proportion of NK cells, which are highly cytotoxic compared to peripheral blood NK cells. NK cells are part of the innate immunity and mediate their cytotoxicity by secreting cytokines and via direct cell-to-cell contact. One of the dominant apoptotic pathways utilised by NK cells is the TNF-α-related apoptosis-inducing ligand (TRAIL) to TRAIL receptor pathway. It has been discovered that pancreatic β-cells express the TRAIL receptor. TRAIL-mediated islet destruction by NK cells was partially but significantly inhibited following the administration of anti-TRAIL mAb in mice. The inhibitory effect was more profound following the co-administration of anti-TRAIL mAb and concanamycin A, an inhibitor of perforin-mediated cytotoxicity [22].

NK cells are traditionally believed to not differentiate into memory cells (cells that have the ability to remember a past encounter with a foreign antigen and stimulate an efficient and enhanced response when re-encountering the same antigen). This is a pathognomonic feature of both T cells and B cells in the adaptive immune system. There is now accumulating evidence that NK cells may possess memory-cell-like properties. NK cells specific to the liver in particular exhibit memory-like properties and responses and lack a classical NK marker DX5, the a2 integral chain CD49b. TNF-α induces the activation of DX5^−^ NK cells through interactions with both TNF-α 1 and 2 receptors on DX5^−^ NK cells. The blocking of TNF-α 1 and 2 receptors on DX5^−^ NK cells with anti-TNF-α antibody treatment prior to transplantation protects islets from NK cell attack during IBMIR. The memory-like liver-resident DX5^−^ NK cells significantly expand in number after primary syngeneic islet transplantation in mice and may target both the originally engrafted primary and secondary transplanted islets. Anti-TNF-α antibody treatment also significantly inhibited the expansion of DX5^−^ NK cells and the prolonged CD69/TRAIL expression on liver NK cells after sequential islet transplantations [23].

NF-κB inhibitors can mitigate the effects of IBMIR on human islets in vitro [24], partly due to their regulation of tissue factor expression. The overexpression of the integral membrane ectonucleotidase CD39 on islets enhances ATP degradation, and it has been shown to limit platelet activation and coagulation without effecting glucose metabolism [25]. The use of complement inhibitors—a C5a inhibitory peptide in a rodent model [26] and a complement inhibitor in a xenotransplant model [27]—showed improved graft survival.

Islets are particularly susceptible to damage mediated by the pro-inflammatory cytokines interleukin (IL)-1, interferon-γ and tumour necrosis factor (TNF)α [28] through NK-κB signalling, the activation of mitogen-activated protein kinases (MAPKs) and Fas-triggered apoptosis. Macrophage-mediated IL-1β secretion may represent the final common pathway for functional islet impairment and destruction through pro-inflammatory cytokine stimulation [29] by enhancing inducible nitric oxide synthase activity and impairing glucose-stimulated insulin secretion [30,31]. IL-1β gene expression was significantly upregulated in in vitro cultures of islets [32], which was exacerbated by serum deprivation and mitigated following incubation with an IL-1 receptor antagonist [33]. Post-transplantation, IL-1β is detectable even in syngeneic grafts [34].

The TNF decoy receptor etanercept has been shown to reduce inflammation and oxidative stress in islets [35]. However, its incorporation into clinical practice proceeded with limited clinical data after being trialled in eight patients [36]. The attractiveness of anti-inflammatory therapeutics to limit IBMIR has persisted, and drug repurposing studies using the IL-1 receptor antagonist anakinra demonstrated improved marginal mass islet engraftment by limiting apoptosis [37]. The combined use of etanercept and anakinra (an IL-1 inhibitor) in allogeneic and autologous islet transplantation demonstrated both safety and tolerability [38,39,40], although larger clinical trials are required to show definitively improved clinical outcomes.

CD47 is a universally expressed cell membrane receptor that ligates signal regulatory protein (SIRP), particularly the alpha domain, to monitor self (versus non-self) and oversee the don’t-eat-me signal. The interaction of parenchymal cell CD47 with myeloid-based SIRPα places it as a checkpoint of innate, allogeneic and xenogeneic immunity. Enhanced CD47 expression correlates with improved solid organ engraftment. The generation of a chimeric CD47-SIRPα protein (containing binding and signal transduction domains) linked to streptavidin was able to limit the macrophage-based phagocytosis of biotinylated cells [41]. Mouse islets engineered to express this protein were protected from the development of an IBMIR-type reaction in vitro and in vivo, facilitating engraftment and long-term syngeneic graft function. These findings were associated with decreased inflammatory cell infiltrates, particularly CD11b^+^Ly6C^hi^/CD11b+/Ly6C^int^ inflammatory monocytes and CD11b^hi^Gr1^hi^ neutrophils.

## 4. Alloimmunity and the Risk of Islet Transplant Failure

Allogeneic islet transplantation requires immunosuppression to limit the rejection of islets, with the potential added benefit of suppressing autoimmunity. Patients with T1DM and a concurrent burden of autoimmune antibodies have a lower rate of islet transplant success due to the presence of memory CD4+ and CD8+ T cells that are rapidly reactivated to target islet antigens (IA-2, GAD-54 and ZnT8) and destroy β-cells [42,43]. Despite their clear prognostic role in the development of T1DM [44], the association between autoantibodies and long-term islet allograft function has been difficult to demonstrate robustly due to small patient numbers. Indeed, patients with T1DM receiving adequate (calcineurin inhibitor and mammalian target of rapamycin inhibitor-based) immunosuppression following islet transplantation demonstrated lymphopenia and a concurrent chronic elevation of IL-7 and IL-15 that promoted T-cell turnover and the expansion of auto-antigen-specific T cells [45].

Human islets bind complement proteins, particularly IgG, IgM, C1q and C3b/iC3b (18431241), leading to lysis and the release of c-peptide. This may be crucial in bridging innate (IBMIR-based) and allogeneic (HLA-based) immune responses in islet transplantation, as C3 can trigger rejection in pre-clinical models of solid organ transplantation [46] and humans [47]. The presence or the development of alloimmunity (to human leukocyte antigens, HLAs) and its effect on allograft survival is well-defined in the solid organ transplantation literature. Donor-specific antibodies (DSAs), pre-formed or de novo, are a leading cause of graft failure due to the development of antibody-mediated rejection in the kidney [48], heart [49] and lung [50], which is associated with graft dysfunction and poorer long-term graft survival. The introduction of molecular typing and Luminex technology has facilitated our ability to precisely define HLAs, the presence (or absence) of antibodies and their (relative) abundance. The risk factors for developing de novo DSAs encompass inadequate immunosuppression (including nonadherence) and inflammation within the graft (rejection) or systemically (infection), which can incite graft immunogenicity and/or heterologous immunity [51]. As many islet transplant patients receive grafts from two–three HLA mismatched donors, the presumed risk of allosensitisation is higher. DSAs binding to endothelial cells or islets (that constitutively express class I and aberrantly upregulate class II HLAs [52]) can activate the classic complement pathway. Even in the absence of complement, some DSAs can promote antibody-dependent cytotoxicity, and innate immune cells bind Fc fragments that trigger degranulation and the release of lytic enzymes from neutrophils and NK cells. C4d, the degradation product of the classical complement pathway, binds covalently to the endothelium and can be used as an immunological marker of antibody-mediated rejection. However, the breadth of islet dispersal throughout the liver parenchyma increases the risk–benefit ratio of a liver biopsy to provide histological assistance for the diagnosis of rejection.

The association of DSAs with islet transplant failure is not as well characterised. It is not known whether the hepatic location of islets is relatively protective given the tolerogenic environment provided by the liver [53]. The prevalence of pre-formed DSAs in islet transplant recipients has been shown to be similar to that of other solid organ transplant cohorts [54], with the suggestion that pre-existing IgM antibodies against HLA class II are associated with improved outcomes. De novo DSAs have been shown to be predictive of islet graft failure [54,55], particularly the development of class I HLAs [56], although this has been disputed [57,58]. Indeed, allogeneic islets may be resistant to DSA-mediated rejection [58] despite the susceptibility with direct binding in vitro, and this is directly due to the endothelial sequestration of DSAs in neo-vascularised islets.

## 5. Non-Immunological Causes of Islet Death

The chronic attrition of islet allograft function over time, despite initial engraftment success, is multifactorial and contributed to by rejection, chronic fibrosis within a non-physiological environment and the drug-induced toxicity of immunosuppression. Both tacrolimus and sirolimus, which are part of standard immunosuppression protocols [3,59], have diabetogenic properties. Calcineurin inhibitors (tacrolimus and ciclosporin) act by limiting the dephosphorylation and translocation of the nuclear factor of activated T cells (NFATs). Calcineurin signalling is required for insulin secretion and β-cell proliferation [60], and the specific inactivation of calcineurin in β-cells is associated with hyperglycaemia with increasing age [61]. Tacrolimus has been shown to increase blood glucose and reduce the homeostasis model assessment of β-cell function (HOMA-β) and the insulin sensitivity index in animals with intact native endocrine pancreatic function [62], following transplantation with human islets [60] and following solid organ transplantation [63]. The effects of short-term tacrolimus exposure promoting hyperglycaemia and compensatory hyperinsulinemia transition to the pseudo-normalisation of insulin, indicative of the loss of insulin secretory capacity, with evidence of β-cell death [64]. Islet apoptosis associated with calcineurin inhibition is also thought to occur by limiting the cAMP response element binding protein (CREB) [65], which decreases IRS-2 expression, limits the phosphorylation of Akt and impacts insulin secretion [66]. CNIs also reduce the cell surface expression of GLUT4 and decrease insulin-stimulated glucose uptake in adipocytes [67], which potentially contributes to peripheral insulin resistance. Tacrolimus promotes a decrease in mitochondrial Ca2+ uptake, which has been shown to impair respiration and ATP production, leading to compromised glucose-stimulated insulin secretion (GSIS) [68]. CNIs, tacrolimus in particular, potentiate the deleterious effect of glucolipotoxicity on β-cells, inducing nuclear FoxO1 expression (which, in turn, limits proliferation [69]) and reducing insulin content and secretion [70].

The incidence of post-transplant diabetes mellitus in solid organ transplantation is the highest in tacrolimus-treated recipients [71,72]. However, mTOR inhibitors are not innocuous in terms of diabetogenic capacity, although much of the literature derives from clinical studies in solid organ (kidney transplant) recipients. An analysis of >20,000 patients in USRDS revealed that kidney transplant recipients without DM receiving sirolimus as part of their immunosuppression regimen were most likely to have Medicare billing for post-transplant DM [73], and the highest HR associated with post-transplant DM was sirolimus and calcineurin inhibitors combined. Tacrolimus and sirolimus both induce reversible graft dysfunction, characterised by amyloid deposition and macrophage infiltration in transplanted islets [60], but without evidence of frank β-cell death. An ultrastructural examination of grafts demonstrated decreased insulin granules, and an accompanying genomic analysis revealed transcripts associated with extracellular matrix deposition and inflammation.

Insulin signalling and β-cell proliferation/survival require intact mammalian target of rapamycin (mTOR), particularly mTORC1 function, which occurs via the insulin receptor substrate 1-Akt-mTOR pathway, where the final step is the phosphorylation of p70 ribosomal protein S6 kinase. In rodent studies, the administration of sirolimus worsened hyperglycaemia, abolished the hyperinsulinemic response and decreased muscle insulin sensitivity in diabetic animals [74]. The latter effect is mediated by glycogen synthase 3β activity [75], and further work has demonstrated that the sirolimus-based dephosphorylation of Yin Yang 1 in skeletal muscle limits insulin signalling [76,77]. Sirolimus has been shown to cause islet death [78] and impair proliferative cell recovery [79]. mTORC2 is required for the insulin-mediated suppression of hepatic gluconeogenesis [80], which is disrupted by sirolimus at higher doses. Not all studies concur with sirolimus impacting insulin action and glucose homeostasis, and this may be related to overall drug exposure at concentrations that affect both mTORC1 and 2.

Immune cell infiltration [81] and amyloid deposition [82] have both been described in liver biopsy results following intraportal transplantation, but these correlate poorly with clinical phenotype. The transplantation of islets from islet amyloid polypeptide (IAPP)-expressing transgenic mice demonstrated early amyloid deposition post-transplantation, reduced β-cell volume and graft failure. This was thought to be due to apoptosis and a concurrent reduction in β-cell replication [83], and both phenomena have been observed in vitro in response to amyloid fibrils. Although IAPP is secreted by β-cells, the human form can aggregate to form cytotoxic fibrils [84]. Interestingly, heparin, which is used to reduce the cyto-destructive effect of IBMIR on islets, promotes the fibrillogenesis of human IAPP, and it has been shown to simultaneously promote amyloid deposition and decrease β-cell apoptosis. Heparinase treatment significantly reduced amyloid deposition and subsequent β-cell cytotoxicity [85].

## 6. The Importance of Islet–Endothelial Crosstalk

Islets are three-dimensional structures, occupying only 2% of the total pancreatic volume but intrinsically linked to abundant vasculature. The islet vascular network comprises 7–8% of the total islet volume and, overall, receives approximately 10× more blood than the surrounding exocrine pancreas [86]. Intact, this capillary system is critical for islet survival and function, and it is central to the secretion of insulin into the circulation for subsequent systemic distribution. Intra-islet endothelial cells (EC) are highly fenestrated. Islet-EC crosstalk provides a number of paracrine effects that promote angiogenesis and β-cell survival, including vascular endothelial growth factor [87], angiopoietins [88], ephrins [89,90] and insulin [91]. The basement membrane also provides a barrier function, structural support and signalling moieties for cellular integrity. The process of islet isolation for transplantation strips β-cells of their vascular infrastructure, and islets are therefore devoid of endothelial cells to support neoangiogenesis, an absolute requirement for engraftment. The basement membrane is also lost during isolation, particularly laminin and collagen [92], and matrix detachment promotes apoptosis [93]. The recovery of structure and function is seen in syngeneic mouse models of islet transplantation, and this is incomplete in allotransplantation [94]. Islets make little matrix and are dependent on endothelial cell recruitment (donor and recipient) for basement membrane repair.

Islets are clusters of approximately 2000 β-cells. Due to the loss of blood flow, the availability of oxygen and nutrients becomes diffusion-dependent, and isolated islets are hypothesised to become hypoxic typically at the central core; however, formal quantitation is difficult. Previous studies have suggested that smaller islets produce superior outcomes [95,96] as a surrogate marker of islet survival. Studies have shown that approximately 70% of transplanted islets remain hypoxic 1 month post-transplantation [97]. However, the exposure of isolated islets to hyperoxic conditions paradoxically worsens the damage of cells residing on the periphery, despite a reduction in central necrosis [98].

The liver is currently considered to be the most suitable site for islet transplantation because, compared to other vascularised beds, such as the spleen and kidney, it is easily accessible with minimal invasion and is (potentially) the least immunogenic. The liver provides a significantly lower oxygen tension (5–10 mmHg) than native pancreatic tissue (30–40 mmHg); however, it allows for a better dispersal of oxygen and nutrients from hepatic sinusoids to transplanted islets. The root of the cause of low revascularisation and oxygen deficiencies in transplanted islets may be intrinsic, as transplanted islets have been reported to have low nitric oxide production, which is essential for regulating vascular tone and blood flow, as well as mitochondrial oxygen consumption. Furthermore, the physiological route of secreted insulin from the pancreas is first directed to the liver via the hepatic portal vein, where it is first consumed before being dispersed to adipose tissue and skeletal muscles [99,100].

β-cells require large amounts of oxygen to meet mitochondrial respiration demands and to facilitate efficient insulin secretion. Hypoxic stress is a contributing factor to cellular dysfunction, islet death during isolation and low islet survival post-transplantation. High numbers of islets are therefore required at transplantation to compensate for cell death and loss. Hypoxia activates the HIF-1α cascade in islets following initial procurement and isolation. HIF-1α accumulates in islet grafts [101], which presumably drives an adaptive response to hypoxia, inducing the expression of pro-angiogenic factors, such as VEGF, AngptI4, Pgf and Anxa2, as well as downstream effectors that initiate angiogenesis [102]. The protective effect of HIF-1α was shown by a poorer survival of transplanted null islets; iron chelation using desferrioxamine increased HIF-1α expression (as well as concurrent ATP content and glucose oxidation) leading to improved islet transplant outcomes [103]. HIF-1α also drives impaired glucose-stimulated insulin secretion [104] and apoptosis [101], an effect mediated by metabolic switching to glycolysis and reduced ATP generation [105]. This has been confirmed in microarray analyses of human isolated islets, where a range of upregulated hypoxia-response genes were identified [106]. The transplantation of hypoxic islets demonstrated a defect in (but not the absence of) β-cell function, which was a consequence of stabilised HIF-1α. HIF-1α activation also initiates apoptosis and mitochondrial caspase-mediated cell death pathways [106], a loss of glucose-stimulated insulin secretion [101] and pro-inflammatory cytokine activation and release, followed by the structural defragmentation of islets. This is exacerbated by significant delays in revascularisation post-transplantation.

The mechanisms of hypoxia-induced β-cell death and dysfunction have been widely investigated, and they are key to facilitating post-transplantation survival. Pancreatic β-cells exposed to acute hypoxia induce caspase-3-mediated apoptosis and endoplasmic reticulum (ER) stress [107]. Recent studies have alluded to ER stress as a causal factor in β-cell dysfunction and islet transplant failure [108]. β-cells are highly metabolic and have a highly developed ER involved in post-translational modification and the assembly and folding of newly synthesised proteins (e.g., insulin in β-cells). An overload of protein production in cells can lead to ER stress. A clinical feature of islet allograft failure (as well as T2DM) is the overproduction of insulin to meet high demands in the context of significant β-cell loss. Insulin overproduction results in protein misfolding and the induction of ER stress through the activation of the C/EBP homologous protein (CHOP), X-box binding protein 1 (XBP-1) and immunoglobulin heavy chain (BIP). To re-establish cellular homeostasis, the adaptive unfolded protein response (UPR) is triggered via three main sensors: activating transcription factor (ATF) 6, double-stranded RNA-dependent protein kinase (PKR)-like ER kinase (PERK) and inositol requiring kinase 1 (IRE1). The UPR responds to ER stress by inducing the upregulation of genes encoding ER chaperone proteins to mitigate protein aggregation and accumulation and to initiate the proteasomal degradation of misfolded proteins. However, in the context of chronic ER stress, the cytoprotective role of the UPR fails, and apoptotic pathways are initiated to induce cell death.

The ER stress marker CHOP is also a pro-apoptotic transcription factor. In response to acute hypoxia, CHOP is upregulated in β-cells. Upon the silencing of CHOP, hypoxia-induced apoptosis is prevented [106]. In addition to environmental hypoxia, intracellular hypoxia is also present in isolated islets. Despite culturing islets in normoxic conditions following isolation, intracellular hypoxia remains. Studies have found that HIF-1α remains overexpressed in isolated islets and that CHOP is upregulated within 4 h of isolation, suggesting early apoptosis. Hypoxia-mediated apoptosis is also implicated in the pathogenesis of T2DM, as pancreatic islets obtained from various murine models (including db/db, ob/ob and kky mice) were discovered to be significantly hypoxic and overexpressed ER stress markers compared to non-diabetic counterparts [106,109].

Chronic hyperglycaemia, which is a clinical feature of both DM and islet allograft failure, damages islets further by triggering β-cell de-differentiation, ER-stress and a loss of function. This phenomenon is known as glucotoxicity. The hyperglycaemic environment in diabetic islet transplant recipients has been implicated in the early loss of transplanted islets. The elimination of glucocorticoids from the immunosuppressive regimen was a significant contributing factor to the success of the Edmonton protocol [59]. One study recently investigated the effects of in vivo chronic hyperglycaemia on ER stress and UPR gene expression in transplanted mouse islets [110]. Diabetic recipients receiving a suboptimal islet transplant that failed to restore euglycemia demonstrated a significant reduction in UPR gene expression, including PERK and IRE1/ATF6, in transplanted islets compared to non-diabetic mice receiving the same β-cell mass. The recovery of adaptive UPR gene expression was observed in diabetic mice transplanted with a sufficient islet mass that established normoglycaemia.

Glucotoxicity is also known to exacerbate other stressors within the graft environment, including inflammation, oxidative stress, hypoxia and impaired vascularisation. Hypoxia decreases ER-to-Golgi protein trafficking and induces cell death by inhibiting the adaptive UPR [111]. Hypoxia mediates these responses independent of HIF-1α activation via several effectors of ER stress, including CHOP, DNA-damage inducible transcript 3 (DDIT3) and c-Jun N-terminal kinase (JNK). Islets isolated from diabetic mice have decreased expressions of adaptive UPR genes, such as *Hspa5*, spliced *Xbp1* and *Fkbp11*. The overexpression of *Hspa5* was found to protect islet β-cells from hypoxia-induced cell death. The silencing of CHOP or JNK also restored UPR gene expression and protein trafficking, providing protection against apoptosis induced by hypoxia [111].

## 7. Future Directions in Islet Transplantation

Emerging strategies to overcome the current limitations of islet transplantation and to improve their success include the encapsulation of islets, the co-transplantation of islets with endothelial cells, stem cell sources for islet transplantation, genetic manipulation and immunosuppression strategies. To mitigate the effects of IBMIR, extrahepatic locations, including the kidney capsule, the spleen, the omental pouch and subcutaneous sites, have been explored in pre-clinical [112,113,114] and human [19] studies. The subcutaneous site is minimally invasive and easily accessible; however, the main limitation is poor vascularisation [115]. Multiple strategies have been developed to counteract this problem and improve the outcomes of transplanted islet survival, including the bioengineering of functional cell sheets using bone-marrow-derived mesenchymal stem cells (MSCs) [116]. MSCs are a source of pro-angiogenic and anti-apoptotic cytokines, including VEGF, HGF, IL-6 and TGFβ1. TGFβ1 triggers the production of heat shock protein 32 (HSP32) and X-linked inhibitor of apoptosis protein (XIAP), which protect islets by supressing oxidative stress [117], inflammation and β-cell apoptosis [118,119]. MSCs can also differentiate into endothelial cells to facilitate peri-islet vascularisation. When engineered into cell sheets, they provided an ideal substrate for human islets and the extracellular matrix that improved islet function and survival [116]. The co-transplantation of islets with adipose-derived mesenchymal stem cell sheets into the subcutaneous site normalised blood glucose levels in diabetic pig recipients [120].

Human dermal fibroblasts have also been studied in vitro as an alternative source to bone-marrow-derived MSCs, as they can be easily harvested from the skin and are highly proliferative. Fibroblasts are also a source of pro-angiogenic factors, such as VEGF and fibroblast growth factor, both of which can improve vascularisation and islet viability post-transplantation. Furthermore, when engineered into cell sheets, fibroblasts maintain the natural structural integrity of islets while also improving their function and survival in vitro [121].

Overcoming insufficient revascularisation to facilitate islet engraftment has been a substantial challenge clinically. The increasing evidence of molecular crosstalk between intra-islet endothelial cells and β-cells [122] suggests that the co-transplantation of these cells may be advantageous. In addition to their contribution to angiogenesis, intra-islet endothelial cells produce various endocrine factors, including thrombospondins, hepatocyte growth factors, collagen and laminins, which promote β-cell survival and improve insulin secretion. Similarly, intra-islet endothelial cells benefit from proliferative factors secreted by β-cells, including ephrins, VEGF, angiopoietins and insulin. Bone-marrow-derived endothelial progenitor cells co-transplanted with islets in diabetic mice have shown promising results in restoring euglycemia through rapid revascularisation [123,124,125]. This was also associated with a strong downregulation of PECAM-1, which is involved in mediating inflammatory responses by promoting the trans-endothelial migration of monocytes, NK cells and neutrophils [123].

Alternative vascularisation units, such as adipose-tissue-derived microvascular fragments (MVFs), may be more efficient at reassembling into microvessels in the initial post-islet-transplantation phase [126,127]. MVFs are also composed of substantial numbers of mesenchymal stem cells, which are known to secrete angiogenic factors and reduce inflammation, thereby protecting islets from hypoxia-induced death. In pre-clinical studies, MVFs co-transplanted with islets under the kidney capsule, as well as the subcutaneous space, were found to improve islet engraftment and restore normoglycaemia in recipients with diabetes [127]. Plasma insulin levels post-transplantation were similar to non-diabetic healthy controls, which was accompanied by enhanced angiogenesis in the grafts. Engineered vascularised organoids have also been shown to re-assemble into larger interconnected channels for perfusion [128], which has been applied to islet transplantation for integration with recipient vasculature [129].

To bypass immune attack following islet transplantation, transplanted islets can be encapsulated in biomaterials (well-reviewed in [130]), including alginate, a polysaccharide derived from seaweed with hydrophilic and biocompatible properties. However, foreign-body responses and fibrotic overgrowth have limited long-term islet viability and graft function [131,132]. Chemical modifications [133] or the addition of anti-TNFα to capsules [134] can improve viability. 

Cell subsets of myeloid, mesenchymal and T cell lineage have the capacity to regulate immune responses and show po-tential as adjuvant immunosuppressive agents in pre-clinical and clinical studies [135]. The intrahepatic infusion of autologous bone-marrow-derived mesenchymal stem cells co-transplanted with islets improved glycaemic control, islet engraftment and quality of life in patients with total pancreatectomy [136]. However, there are no current clinical trials investigating cell therapy in allogeneic islet transplantation.

## 8. Conclusions

Islet transplantation is an effective treatment that reduces severe hypoglycaemia, re-establishes hypoglycaemia awareness, stabilises glycaemic control and can provide insulin independence. Its long-term benefits also include reduced morbidity from microvascular complications, particularly retinopathy and nephrotoxicity [137]. The benefits of islet transplantation are balanced by the need for immunosuppression—and the concordant risks of malignancy, infection and cardiovascular disease that manifest with long-term exposure to immunomodulatory agents—and the relatively ephemeral duration of successful islet transplantation (only 40% of patients achieve long-term insulin independence). The failure to meet graft survival outcomes similar to those of solid organ transplantation may be due to the susceptibility of β-cells to demise. This effect is multifactorial, but the development of strategies to improve β-cell durability or to reverse the adverse changes that are initiated by the islet isolation process will improve the success and duration of islet transplantation. The use of alternative sources of islets—xenotransplantation or stem-cell-derived sources—may help to reverse the current imbalance between supply and demand, but much remains unknown regarding their immunogenicity and physiological function. Further refinements in immunosuppressive drug regimens are also required to limit the harmful effects on both β-cells and remote organs.

## Figures and Tables

**Figure 1 cells-12-00796-f001:**
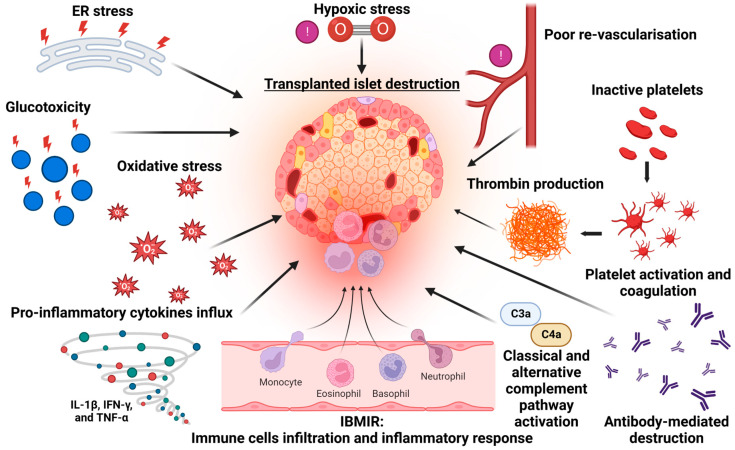
Mechanisms of beta cell destruction following islet transplantation.

## Data Availability

Not applicable.

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
