# Peer review of "No Time to Die—How Islets Meet Their Demise in Transplantation"

_cells, 2023, doi:10.3390/cells12050796_

Round 1

Reviewer 1 Report

11.    The authors noted that Type 1 DM (T1DM) is characterized by auto-immune destruction of b-cells, although the notion of b-cell complicity in their own destruction due to limited responses to survive an inflammatory insult, is touted as a competing hypothesis for disease development. This is an interesting remark by the authors. It would be nice if they can link this statement to the emerging role of defensins in the etiology of T1DM.  

22. Some studies have suggested that the incorporation of oxygen generating particles such as sodium percarbonate into the isolation medium may be used as a strategy to diminish hypoxia during the islet isolation process and thus improve the success and duration of islet transplantation. The authors need to include this observation.

Author Response

The authors noted that Type 1 DM (T1DM) is characterized by auto-immune destruction of b-cells, although the notion of b-cell complicity in their own destruction due to limited responses to survive an inflammatory insult, is touted as a competing hypothesis for disease development. This is an interesting remark by the authors. It would be nice if they can link this statement to the emerging role of defensins in the etiology of T1DM.  

 Author response: After modifications to the text we are now at >4900 words. We feel that text on the emerging role of defensins in T1DM is outside the scope of this review, particularly given there are no data on its role in islet transplantation.

Some studies have suggested that the incorporation of oxygen generating particles such as sodium percarbonate into the isolation medium may be used as a strategy to diminish hypoxia during the islet isolation process and thus improve the success and duration of islet transplantation. The authors need to include this observation.

Author response: We have included additional text around emerging strategies that are islet protective.

Reviewer 2 Report

The authors of the manuscript " No Time to Die: how islets meet their demise in transplantation" have effectively compiled the series of events that lead to islet cell destruction post-transplant. However, some of the events such as the role of complement in islet cell loss have been superficially discussed. Similarly, the role of immunosuppression-mediated islet loss has also been mildly touched upon but lacks mechanistic details. Overall, the review requires minor additions to this process of cell destruction.  

Author Response

The authors of the manuscript " No Time to Die: how islets meet their demise in transplantation" have effectively compiled the series of events that lead to islet cell destruction post-transplant. However, some of the events such as the role of complement in islet cell loss have been superficially discussed. Similarly, the role of immunosuppression-mediated islet loss has also been mildly touched upon but lacks mechanistic details. Overall, the review requires minor additions to this process of cell destruction. 

Author response: We have included additional text on complement-mediated islet cell loss and mechanisms underlying immunosuppressive drug toxicity

Reviewer 3 Report

This review adequately discusses the limitations to clinical islet transplantation. No changes needed. 

1. This was a review article - not research. Good overview, with some elements of more speculative parts, like the NK memory cells role in islet loss. 2. Topic is relevant. Again, this is a review article. They covered the known pertinent issues affecting islet dysfunction. One paragraph could be added about toxicity to other drugs than steroids. FK and CsA are both known to be diabetogenic - but CNI remain backbone in most IS regimens. They could discuss non-CNI IS, like anti-CD40L, which has been in development for the last 25 years. 3. Review paper - obvious overlap with published material. 4. Review paper. No methods are asked for. No controls. No experiments. 5. Conclusions are adequate. No issues. 6. References are appropriate. 7. Figures are clear and informative.

Author Response

Author response: We would like to thank the reviewer for their remarks. We have discussed islet-based side effects in immunosuppressive drugs used clinically (including CNI and non-CNIs).

Reviewer 4 Report

The review by Kale and Rogers contains a well written and well organized viewpoint regarding islet viability and function after transplantation. A few points, if addressed, would significantly improve the clarity of the manuscript and increase its impact on the field. Those points are as follows:

1) The authors state that beta-cell function after transplantation resembles failure of islet function in T2DM (lines68-71). However, no details were given to explain how that similarity might be true (e.g., morphologically, gene expression, etc.). The authors should clarify this point and include references or remove the statement about resembling T2DM.

2) In line 80, two possible death mechanisms were provided (apoptosis and/or pyroptosis) but necrosis was left out. Necrosis should be included or explained why it doesn’t occur. Also references should be provided to support or refute the involvement of each death pathway or statement revised/removed if no evidence exists.

3) In lines 138-140, the authors state that incorporation into clinical practice proceeded with limited clinical data. Do they mean clinical efficacy was lacking or that any supporting data at all was limited? This point should be clarified. If there were shortcomings in the clinical data, please outline them.

4)In lines 206-208, the authors should replace the book chapter with the primary reference(s) that support the point about blood flow into islets relative to exocrine tissue.

5) Figure 1 would be more informative if the authors split it into A) and B) style panels where immune-mediated events versus non-immune mediated events could be delineated more clearly.

Author Response

Author response: We would like to thank the reviewer for their comments.

We feel strongly about the statement that islet transplant failure mimics beta cell failure in T2DM. We have left this in the text (page 5) and highlighted relevant mechanisms that are identical in both disease processes.

The process of cell death in IBMIR likely overlaps the range of frank necrosis, pyroptosis, necroptosis etc. Most pre-clinical studies describing this phenomenon have used TUNEL staining which detects generic cell death, and no study has investigated differing mechanisms in significant detail. We have revised the text on page 5 to reflect this lack of information.

With regard to the comment about “limited clinical data” this was referring to the few patient numbers in this study. We have modified the text to reflect this.

Thank you for pointing out the reference discrepancy – we have made this change.

 We have opted to leave Figure 1 it as is.